# Further evidence in support of the validity of the post-secondary student stressors index using a nationwide, cross-sectional sample of Canadian university students

**Brooke Linden** [1,2] *, **Heather Stuart** [1,2]

**1** Department of Public Health Sciences, Queen's University, Kingston, Ontario, Canada, **2** Health Services and Policy Research Institute, Queen's University, Kingston, Ontario, Canada

* brooke.linden@queensu.ca

## Abstract

The Post-Secondary Student Stressors Index (PSSI) was created to facilitate improved evaluation of the sources of post-secondary student stress. This study reports evidence in support of the validity of the tool using a large, nationwide cross-sectional sample of students attending universities across Canada during the 2020–2021 academic year. We provide additional evidence for the construct validation of the PSSI, including internal structure evidence and relations to other variables by estimating multiple-indicator, multiple-cause models and investigating Spearman's rho correlation coefficients between the PSSI and like constructs. Based on index validation guidelines, results provide further support for the internal structure of the PSSI, demonstrating hypothesized relationships with like constructs and manifest variables, as well as acceptable goodness-of-fit statistics. Similarly, correlation coefficients were statistically significant and in line with directionality hypotheses. The results of this research provide further evidence for the validity of the PSSI among varied university student populations in Canada and addresses several of the limitations identified in earlier preliminary psychometric work on the instrument.

## Introduction

Although the prevalence of mental illnesses among post-secondary students are not significantly different from those observed among the general Canadian population [1], the demand for mental health services on post-secondary campuses continues to grow as does the complexity of presenting issues [2]. National cross-sectional data suggests that the prevalence of students reporting above average to extreme levels of stress is increasing, along with self-reported symptoms of common mental illnesses such as anxiety and depression [3]. Chronic stress has been linked to poor academic performance and negative mental health outcomes [4], necessitating the bolstering of upstream campus services that provide students with the tools needed to recognize and respond to changes in their mental health in the face of day-to-day stressors. In order to appropriately tailor these upstream services to the needs of students, a method of measuring the sources of student stress is required.

**Data Availability Statement:** There are ethical restrictions to sharing this data, put in place by the 15 ethics review board(s) from which we received clearance. A stipulation of our ethics approvals

across all sites was that the students' data would not be shared without an additional layer of ethics approval. All secondary analyses require a subsequent ethics request to the Queen's University's Health Sciences and Affiliated Teaching Hospitals Research Ethics Board (HSREB) as the primary ethics body that approved this study. Requests regarding access to the data used in this study can be directed to the HSREB (hsreb@queensu.ca, 613-533-6000 ext. 78223), who will forward these requests to the data custodian (Dr. Brooke Linden, brooke.linden@queensu.ca). We are more than happy to share the data with other investigators, and encourage collaborative analyses.

**Funding:** The authors received no specific funding for this work.

**Competing interests:** I have read the journal's policy and the authors of this manuscript have the following competing interests: BL is the developer of the Post-Secondary Student Stressors Index (a non-financial conflict of interest). HS is the Bell Canada Mental Health and Anti-Stigma Research Chair.

Until recently, there has been no validated measurement tool for assessing the frequency and severity of stressors experienced by students within post-secondary settings. Existing instruments designed to evaluate student stress are associated with a host of issues, including being conceptualized too broadly or too narrowly, seldom involving students in the development process (often focusing on a single year or program), or demonstrating a lack of strong psychometric properties. These instruments are also often outdated, with the majority having been developed between 1990 and 2005. This raises concerns regarding the validity of the instruments with respect to accurate and comprehensive measurement of the stressors that are currently salient for students [5].

In 2019, the Post-Secondary Student Stressors Index (PSSI) was developed in response to these gaps in measurement. The tool was developed and validated with extensive collaboration with students over the course of a two-year period. While a preliminary psychometric analysis was conducted on the instrument in 2019 using pilot test data from a single school in Ontario [6], instrument validation is an ongoing, iterative process [7]. Therefore, the purpose of this study was to collect further evidence of validity (specifically, internal structure and relations to other variables) using a larger, more varied sample, representing university students from across Canada.

## Methods

### Study design

The data analyzed here was collected during the study, "Multi-site Release of the Post-Secondary Student Stressors Index (PSSI)", conducted during the 2020–2021 academic year. Cross-sectional data were collected at three time points via an online survey from students ($n$ = 12,577) attending fifteen Canadian universities [8]. Participating institutions represented all but one Canadian province. The survey, distributed through Qualtrics, included questions about demographic characteristics, sources of student stress, mental health and mental illness, and stress specific to COVID-19. Participants were provided with a letter of information and indicated their consent by selecting "I have read and understood the Letter of Information and agree to participate in this study" before being granted access to the survey questions. The sampling method and sample size used was at the discretion of each participating institution. Further detail related to study design can be found elsewhere [8].

### Measures

**Demographics.**   Several demographic variables were assessed, including: gender (male, female, non-binary), age in years, level of study (undergraduate, graduate, or professional program), marital status, residence during the academic year, student status (full-time/part-time/other), first-generation status (yes/no), international student status (yes/no), self-reported grade point average (GPA), and region of university (Eastern/Western/Central Canada).

**Post-secondary student stressors index.**   The Post-Secondary Student Stressors Index (PSSI) is a 46-item instrument. For each item, respondents are asked to indicate the severity of stress experienced and the frequency with which they experienced it. Responses are provided on an adjectival scale from 1 ('not stressful' and 'rarely') to 4 ('extremely stressful' and 'almost always'), with higher ratings indicating a greater level of stress resulting from an item. An additional option to indicate "not applicable" (a response of 0) was also available in the event that a stressor didn't happen or was otherwise not applicable. The tool evaluates stressors across five domains, including: academics, learning environment, campus culture, interpersonal, and personal. For the purposes of the subsequent analyses only severity ratings were used.

**Mental health measures.**   The Perceived Stress Scale (PSS10) is a brief, 10-item scale designed to measure general stress [9]. Participants are asked to indicate the frequency with which they have had certain thoughts or feelings on an adjectival scale from 0 ('never') to 4 ('very often'). Positively worded items on the scale are reverse coded and ratings then summed, with higher scores indicating more perceived stress. The Kessler Psychological Distress Scale (K10) is a 10-item scale designed to measure general psychological distress [10]. Participants are asked to indicate the frequency with which they have had certain thoughts or feelings on an adjectival scale from 1 ('none of the time') to 5 ('all of the time'). Scores are then summed, with higher scores reflecting greater distress and a higher likelihood of experiencing a mental disorder. Both the PSS10 and K10 have demonstrated consistent psychometric properties in a number of samples and settings and have previously been used among post-secondary populations [11–13].

## Analysis

The statistical tests we used to collect evidence for validity reflect those appropriate for an index, as opposed to a scale. While both types of multi-item instruments are used to evaluate latent constructs (i.e., variables that cannot be directly observed), items in a scale are referred to as "effect indicators", while those in an index are referred to as "causal indicators" [7]. With the latter, changes in the items (stressors) determine changes in the value of the latent variable (student stress). The goal of an index is not unidimensionality and internal consistency. On the contrary, there is no expectation for items within an index to correlate with one another, although they might [14, 15]. Instead, index construction places emphasis on considering multicollinearity among indicators and assessing theoretical relationships between latent constructs and predictor variables [16]. Most importantly, while the goal in scale development is often to use as few items as possible to measure the latent construct in the interest of brevity, the opposite is true of index development. Indices often include a large number of items as the goal is to cover the entire scope of the latent variable, which may be quite broad. In general, items in an index are retained as long as they have a distinct influence on the latent construct, as removing an indicator in an index may result in omitting a part of the construct itself [16, 17].

Therefore, to further evaluate the *internal structure* of the PSSI, we estimated a multiple-indicator, multiple-cause (MIMIC) model, a special case of structural equation modelling (SEM) and an extension of Confirmatory Factor Analysis (CFA) [18]. A MIMIC model utilizes both a psychometric (i.e., measurement) approach as well as a structural approach, which allows for the researcher to estimate relationships among latent constructs (unobserved variables) and contextualize them using predictor variables (usually demographics assumed to have no error, such as age or gender) [18]. We hypothesized that the model would demonstrate that the internal structure of the PSSI was consistent with an index and demonstrate acceptable goodness-of-fit statistics. To assess multicollinearity among indicators, we examined the variance inflation factor (VIF) using a cut-off threshold of 10 [19].

Next, we explored the relationships between the latent variables on the PSSI and other "like" constructs to assess *relations to other variables* [20]. This method of index validation requires that these constructs are measured using effect indicators and that a theoretical relationship can be presumed to exist between the constructs [17]. We hypothesized that higher scores on the PSSI latent variables (indicating higher student-specific stress) would predict higher scores on both the PSS10 and K10 (higher general stress and psychological distress, respectively) resulting in positive, significant estimates. To test this hypothesis, we specified a second MIMIC model including PSS10 and K10 scores as predicted variables. Next, we

calculated nonparametric Spearman's rho ($r_s$) correlation coefficients to examine the relationships between the PSSI, PSS10, and K10 across a variety of student subgroups, expecting to see positive, moderately significant relationships across all groups.

All tests were conducted using a complete case analysis approach to missing data; any student who had complete information for the variables relevant to an individual test was included in the analysis. All analyses were completed using R, Version 3.4.1. This research received ethics clearance from Queen's University's Health Sciences and Affiliated Teaching Hospitals Research Ethics Board.

## Results

### Sample

Descriptive statistics for the demographic characteristics of the sample are reported in Table 1. Most participants were female (71.6%), single (85.6%), lived off campus with family (54.0%), self-reported their GPA to be in the A range (60.3%), and studied full-time (90.6%) at the undergraduate level (75.0%). Most participants were aged 20 years or younger (41.4%), with an overall average age of 23 ($SD = 6.9$). Approximately 12% of the sample were international students, and over one quarter of respondents (26%) were the first in their families to attend university.

### Internal structure evidence

All 46 items on the PSSI were first correlated with the PSS10 in line with recommendations to explore correlations between index indicators and a separate variable that "summarizes the essences of the construct that the index purports to measure" [16, 17]. The PSS10 is a validated measure of general perceived stress, thus fulfilling this requirement. We assessed the potential for multicollinearity by examining the variance inflation factor (VIF) for each indicator. All VIF values were well below the cut-off ($<5$).

As previously outlined, the PSSI was conceptualized as being an index composed of five domains (or latent constructs) of student stress: academics, learning environment, campus culture, interpersonal, and personal stressors. As such, we specified these five latent variables in our MIMIC model and included manifest variables age and gender (where 1 = female and 0 = male) as predictors for our structural model. Descriptive statistics for indicators and covariates are shown in Table 2. We decided to run two models. In the first model, we excluded observations where a participant indicated that a stressor was "not applicable" or did not happen. This resulted in a model with 516 total observations. In the second model, we included those observations, resulting in a model with 11,965 total observations. Results of both models are displayed in Table 3.

In the first model, gender was significantly associated with all latent constructs, while age was significantly associated with all but the "learning environment" and "personal". All items demonstrated strong factor loadings ($>0.5$). The model was statistically significant ($X^2 = 4218$, $df = 1061$, $p<0.001$) and demonstrated good overall fit. The root mean square error of approximation (RMSEA) and standardized root mean square residual (SRMR) were both $<1$, and both the Comparative Fit Index (CFI) and Tucker-Lewis Index (TLI) were only just below the desired cut-off of $\geq 0.90$ at 0.81 and 0.80, respectively. In contrast, the second model revealed several low (and in one case, negative) factor loadings and weaker overall fit statistics. The lower factor loadings were observed on items where a much larger proportion of respondents indicated that the stressor was "not applicable" or did not happen to them relative to the other items on the index. In this model, gender and age were significantly associated with all latent constructs, though this is to be expected given the large sample size. Similarly, though the

**Table 1. Demographic characteristics of sample ($n$ = 12 577).**

| Variable | n | % |
|---|---:|---:|
| **Gender** | | |
| Female | 9007 | 71.6 |
| Male | 3195 | 25.4 |
| Non-binary | 238 | 1.9 |
| Prefer not to answer | 136 | 1.1 |
| **Age** | | |
| 20 years and under | 5205 | 41.4 |
| 21–24 years | 3949 | 31.4 |
| 25–29 years | 1724 | 13.7 |
| 30 years and over | 1683 | 13.4 |
| **Level of Study** | | |
| Undergraduate | 9437 | 75.0 |
| Graduate | 2367 | 18.8 |
| Professional Program | 466 | 3.7 |
| Other | 260 | 2.1 |
| Prefer not to answer | 47 | 0.4 |
| **Marital Status** | | |
| Single[1] | 10768 | 85.6 |
| Married or common-law | 1560 | 12.4 |
| Separated, Divorced, or Widowed | 112 | 0.9 |
| Prefer not to answer | 137 | 1.1 |
| **Residence During Academic Year** | | |
| On campus residence/other | 965 | 7.6 |
| Off campus with roommates | 3346 | 26.6 |
| Off campus alone | 1267 | 10.1 |
| Off campus with family | 6794 | 54.0 |
| Prefer not to answer | 211 | 1.7 |
| **Student Status** | | |
| Full-time | 11390 | 90.6 |
| Part-time | 1064 | 8.5 |
| Other | 92 | 0.7 |
| Prefer not to answer | 31 | 0.2 |
| **First Generation Student** | | |
| No | 9247 | 73.5 |
| Yes | 3263 | 25.9 |
| Prefer not to answer | 67 | 0.5 |
| **International Student** | | |
| No | 10962 | 87.2 |
| Yes | 1590 | 12.6 |
| Prefer not to answer | 25 | 0.2 |
| **GPA** | | |
| A | 7580 | 60.3 |
| B | 3608 | 28.7 |
| C | 779 | 6.2 |
| D/F | 61 | 0.5 |
| Prefer not to answer | 542 | 4.3 |
| **Region** | | |

*(Continued)*

**Table 1.** (Continued)

| Variable | n | % |
|---|---|---|
| Atlantic Canada | 4364 | 35.6 |
| Western Canada | 5441 | 44.4 |
| Central Canada | 2445 | 20.0 |

(1) "Single" includes respondents who indicated they were 'single and not dating', 'single and dating', and in a relationship (not married)'

model itself was found to be significant, the chi-square statistic is sensitive to sample size, and thus not informative. Ultimately, results suggest that the removal of responses of "not applicable" or "did not happen" results in a stronger overall model.

## Relations to other variables

To assess evidence for relations to other variables, we first specified a third MIMIC model including PSS10 and K10 scores as predicted variables (Table 4). Estimation of the model resulted in a good overall fit and, as expected, the path between the PSSI latent variables and selected indicators were significant and above 0, consistent with our hypothesis.

Next, we assessed the correlations between the PSSI and like instruments, using a non-parametric Spearman's rho as the data were not normally distributed. Given that it would not be appropriate to sum participants' responses on the PSSI, we created a "count" variable. The goal of the PSSI is not to derive a comprehensive severity "score" based on the stressors experienced, but rather to identify sources of stress. Logically, we would expect that those who experienced a greater number of stressors would in turn experience a higher level of overall stress and/or psychological distress, as measured by the PSS10 and K10. Thus, to develop the PSSI count variable, we dichotomized responses such that any response indicating stress in response to a stressor was coded as 1, and responses indicating that a stressor was "not applicable", did not happen, or was not stressful were coded as 0. Counts were then summed to derive a count. Essentially, this applies an equal weight of 1 to each stressor, a common method employed when working with health status indices [7]. To assess these relationships across different subgroups of students, we repeated the analysis across several split samples. Results are displayed in Table 5. As hypothesized, the analyses revealed moderate, positive correlations across all groups. All correlations were significant at $p < 0.001$.

## Discussion

This psychometric analysis followed the guidelines laid out by Diamantopoulos and Winklhofer regarding validation processes appropriate for an instrument developed during a formative measurement perspective (i.e., an index rather than a scale) [16, 17]. Notably, no reliability analyses nor dimensionality tests, such as exploratory factor analysis, were performed on the PSSI as unidimensionality and internal consistency are not relevant when evaluating the validity of an index [14]. The results of the analyses presented here echo the preliminary analyses previously published [6] and provide further support for the validity of the Post-Secondary Student Stressors Index (PSSI). Results demonstrate its utility among a large sample of students who varied by gender, age, level of study, course load (i.e., part-time vs. full-time), and more.

Unlike other tools that purport to evaluate the sources of student stress, the PSSI was co-developed with students, and has now been validated using a large sample of students

**Table 2. Descriptive statistics for indicators and covariates.**

| | Female (*n* = 9007) | | | Male (*n* = 3195) | | |
|---|---|---|---|---|---|---|
| | *Mean* | *SD* | *% NA* | *Mean* | *SD* | *% NA* |
| **Covariates** | | | | | | |
| Age | 23.4 | 6.8 | – | 23.9 | 7.2 | – |
| PSS10 | 23.2 | 6.1 | – | 20.6 | 6.6 | – |
| K10 | 29.3 | 9.4 | – | 26.3 | 9.4 | – |
| **Indicators** | | | | | | |
| *Academic* | | | | | | |
| Preparing for exams | 2.8 | 0.8 | 8.0 | 2.5 | 0.8 | 5.4 |
| Writing exams | 2.9 | 0.8 | 8.4 | 2.5 | 0.9 | 5.9 |
| Writing multiple exams around the same time | 3.5 | 0.7 | 20.5 | 3.2 | 0.8 | 18.9 |
| Exams worth more than 50% of course grade | 3.4 | 0.7 | 17.8 | 3.0 | 0.9 | 13.6 |
| Heavily weighted assignments | 2.7 | 0.8 | 3.5 | 2.4 | 0.9 | 3.1 |
| Having multiple assignments due around the same time | 3.0 | 0.8 | 3.3 | 2.8 | 0.9 | 3.3 |
| Managing my academic workload | 2.6 | 0.8 | 0.5 | 2.4 | 0.9 | 0.6 |
| Receiving a bad grade | 3.2 | 0.8 | 5.8 | 2.9 | 0.9 | 5.5 |
| Managing a high GPA | 2.9 | 0.9 | 2.9 | 2.5 | 1.0 | 4.0 |
| Working on my thesis | 2.8 | 0.9 | 67.0 | 2.6 | 0.9 | 62.5 |
| Performing well at my professional placement (i.e., practicum, clerkship) | 2.6 | 0.9 | 57.7 | 2.3 | 0.9 | 56.3 |
| *Learning Environment* | | | | | | |
| Poor communication from professor | 2.9 | 0.8 | 5.9 | 2.7 | 0.9 | 5.9 |
| Unclear expectations from professor | 3.0 | 0.8 | 4.3 | 2.8 | 0.9 | 5.1 |
| Lack of guidance from professor | 2.8 | 0.8 | 5.4 | 2.6 | 0.9 | 5.8 |
| Meeting with my professor | 2.0 | 1.0 | 13.5 | 1.7 | 0.9 | 12.5 |
| Meeting my thesis/placement supervisor's expectations | 2.5 | 0.9 | 61.5 | 2.3 | 0.9 | 59.6 |
| Lack of mentoring from my thesis/placement supervisor | 2.6 | 1.0 | 66.9 | 2.4 | 1.0 | 63.7 |
| *Campus Culture* | | | | | | |
| Adjusting to the post-secondary lifestyle | 2.1 | 0.9 | 6.9 | 1.9 | 0.9 | 6.9 |
| Adjusting to my program | 2.1 | 0.9 | 4.2 | 1.9 | 0.8 | 4.3 |
| Academic competition among my peers | 2.1 | 1.0 | 7.5 | 1.8 | 0.9 | 9.4 |
| Feeling like I'm not working hard enough | 2.9 | 0.9 | 2.9 | 2.7 | 0.9 | 3.6 |
| Feeling like my peers are smarter than I am | 2.6 | 1.0 | 4.3 | 2.2 | 1.1 | 6.9 |
| Pressure to succeed | 3.0 | 0.9 | 1.1 | 2.8 | 1.0 | 2.1 |
| Discrimination on campus | 2.0 | 1.0 | 59.5 | 1.9 | 1.0 | 61.8 |
| Sexual harassment on campus | 2.3 | 1.0 | 62.2 | 1.8 | 1.0 | 73.5 |
| *Interpersonal* | | | | | | |
| Making new friends | 2.1 | 1.0 | 7.9 | 1.9 | 1.0 | 8.8 |
| Maintaining friendships | 2.0 | 0.9 | 3.8 | 1.8 | 0.9 | 5.2 |
| Networking with the "right" people | 2.2 | 0.9 | 9.4 | 2.1 | 0.9 | 10.1 |
| Feeling pressured to socialize | 2.2 | 1.0 | 10.0 | 1.9 | 1.0 | 12.6 |
| Balancing a social life with academics | 2.5 | 0.9 | 2.9 | 2.3 | 1.0 | 4.2 |
| Comparing myself to others | 2.7 | 1.0 | 4.1 | 2.3 | 1.0 | 7.5 |
| Comparing my life to others' on social media | 2.3 | 1.0 | 11.4 | 1.9 | 1.0 | 24.9 |
| Meeting other peoples' expectations of me | 2.6 | 1.0 | 4.0 | 2.3 | 1.0 | 8.7 |
| Meeting my own expectations | 3.1 | 0.9 | 0.5 | 2.8 | 1.0 | 1.4 |
| *Personal* | | | | | | |
| Making sure that I get enough sleep | 2.3 | 1.0 | 0.9 | 2.1 | 1.0 | 1.8 |
| Making sure that I get enough exercise | 2.4 | 0.9 | 2.6 | 2.2 | 0.9 | 2.8 |

(*Continued*)

**Table 2.** (Continued)

|  | Female (*n* = 9007) | | | Male (*n* = 3195) | | |
|---|---|---|---|---|---|---|
|  | *Mean* | *SD* | *% NA* | *Mean* | *SD* | *% NA* |
| Making sure that I eat healthy | 2.3 | 0.9 | 1.2 | 2.0 | 0.9 | 2.3 |
| Having to prepare meals for myself | 2.0 | 1.0 | 5.8 | 1.8 | 0.9 | 8.9 |
| Balancing working at my job with my academics | 2.7 | 0.9 | 30.8 | 2.5 | 1.0 | 38.5 |
| Balancing my extracurriculars with academics | 2.2 | 0.9 | 25.4 | 2.1 | 0.9 | 24.8 |
| Feeling guilty about taking time for my hobbies/interests | 2.7 | 1.0 | 4.7 | 2.4 | 1.0 | 7.5 |
| Having to take student loans | 2.8 | 1.1 | 50.7 | 2.5 | 1.1 | 52.1 |
| Worrying about paying off debt | 2.9 | 1.0 | 40.3 | 2.6 | 1.1 | 43.0 |
| Worrying about getting a job after graduating | 2.9 | 1.0 | 7.9 | 2.7 | 1.1 | 9.5 |
| Worrying about getting into a new program after graduating | 2.8 | 1.1 | 29.2 | 2.4 | 1.1 | 32.5 |
| Worrying about reaching major "life events" (i.e., buying a house, marriage) | 2.8 | 1.0 | 8.4 | 2.5 | 1.1 | 12.0 |

(1) PSSI responses range from 1 to 4 where higher scores are equivalent to higher severity of stress.

(2) # NA column indicates the proportion of participants indicating that the stressor was "not applicable" or did not happen. These responses were removed from mean calculations.

attending university across Canada. The PSSI is a valuable tool for universities that are interested in identifying the presence and distribution of student stressors on their campuses. A more in-depth understanding of the sources of student stress, as well as their associated severity and frequency is an important step towards improved tailoring of upstream campus mental health and wellbeing services. Although downstream campus services (i.e., psychotherapy, pharmacotherapy) continue to be overburdened, few post-secondary institutions have successfully taken a comprehensive or "whole campus approach" including both upstream and downstream mental health services, as is recommended by broad frameworks like the Okanagan Charter. Post-secondary institutions are increasingly encouraged to incorporate both physical and mental health into institutional culture and policies to promote an environment that enhances health and well-being, instead of focusing on downstream treatment only [21]. By improving understanding of the sources of student stress and areas most in need of support through the use of tools like the PSSI, institutions may be able to improve the effectiveness of upstream services, thereby contributing toward alleviating the bottleneck currently observed at the downstream service level.

## Limitations

There are some limitations to this research. While the results demonstrate additional evidence for the validity of the tool among students attending schools across a variety of regions in Canada, the sample did not include participants from the province of Quebec. Additionally, the French version of the PSSI has not yet been formally tested. As a result, the findings presented here cannot be generalized to the francophone university student population in Canada. Secondly, the PSSI, though intended for use cross various types of post-secondary institutions, has thus far only been validated for use among university student populations. Therefore, caution should be used when applying this tool among samples of post-secondary students attending colleges, institutes, etc. Finally, the data used for this study were collected during the 2020–2021 academic year, which was the first full academic year that took place during the global COVID-19 pandemic. As a result, it is likely that stress levels (both general and student-specific) were somewhat higher than we might usually expect. However, this would only be expected to impact the descriptive analyses, and not the psychometric analyses.

Table 3. Multiple cause, multiple indicator (MIMIC) model estimates.

| Parameter | Indicators | Latent Variable | Estimates | |
|---|---|---|---|---|
| | | | Model 1 | Model 2 |
| | | | n = 516 | n = 11 965 |
| Factor Loading | Preparing for exams | Academic | 0.717 | 0.793 |
| | Writing exams | | 0.728 | 0.787 |
| | Writing multiple exams around the same time | | 0.702 | 0.678 |
| | Exams worth more than 50% of course grade | | 0.693 | 0.710 |
| | Heavily weighted assignments | | 0.710 | 0.576 |
| | Having multiple assignments due around the same time | | 0.733 | 0.577 |
| | Managing my academic workload | | 0.746 | 0.451 |
| | Receiving a bad grade | | 0.691 | 0.618 |
| | Managing a high GPA | | 0.737 | 0.624 |
| | Working on my thesis | | 0.585 | -0.025 |
| | Performing well at my professional placement (i.e., practicum, clerkship) | | 0.620 | 0.189 |
| | Poor communication from professor | Learning Env | 0.802 | 0.824 |
| | Unclear expectations from professor | | 0.830 | 0.858 |
| | Lack of guidance from professor | | 0.844 | 0.826 |
| | Meeting with my professor | | 0.587 | 0.331 |
| | Meeting my thesis/placement supervisor's expectations | | 0.619 | 0.090 |
| | Lack of mentoring from my thesis/placement supervisor | | 0.705 | 0.185 |
| | Adjusting to the post-secondary lifestyle | Campus Culture | 0.769 | 0.533 |
| | Adjusting to my program | | 0.794 | 0.537 |
| | Academic competition among my peers | | 0.798 | 0.644 |
| | Feeling like I'm not working hard enough | | 0.774 | 0.696 |
| | Feeling like my peers are smarter than I am | | 0.790 | 0.747 |
| | Pressure to succeed | | 0.784 | 0.729 |
| | Discrimination on campus | | 0.505 | 0.305 |
| | Sexual harassment on campus | | 0.489 | 0.303 |
| | Making new friends | Interpersonal | 0.730 | 0.559 |
| | Maintaining friendships | | 0.732 | 0.589 |
| | Networking with the "right" people | | 0.812 | 0.566 |
| | Feeling pressured to socialize | | 0.766 | 0.600 |
| | Balancing a social life with academics | | 0.791 | 0.622 |
| | Comparing myself to others | | 0.810 | 0.765 |
| | Comparing my life to others' on social media | | 0.749 | 0.641 |
| | Meeting other peoples' expectations of me | | 0.803 | 0.700 |
| | Meeting my own expectations | | 0.711 | 0.622 |
| | Making sure that I get enough sleep | Personal | 0.682 | 0.598 |
| | Making sure that I get enough exercise | | 0.739 | 0.654 |
| | Making sure that I eat healthy | | 0.715 | 0.693 |
| | Having to prepare meals for myself | | 0.655 | 0.551 |
| | Balancing working at my job with my academics | | 0.752 | 0.380 |
| | Balancing my extracurriculars with academics | | 0.761 | 0.409 |
| | Feeling guilty about taking time for my hobbies/interests | | 0.744 | 0.581 |
| | Having to take student loans | | 0.662 | 0.411 |
| | Worrying about paying off debt | | 0.686 | 0.433 |
| | Worrying about getting a job after graduating | | 0.692 | 0.504 |
| | Worrying about getting into a new program after graduating | | 0.681 | 0.403 |
| | Worrying about reaching major "life events" (i.e., buying a house, marriage) | | 0.714 | 0.540 |

(*Continued*)

**Table 3.** (Continued)

| Parameter | Indicators | Latent Variable | Estimates | | | |
|---|---|---|---|---|---|---|
| | | | Model 1 | | Model 2 | |
| | | | $n$ = 516 | | $n$ = 11 965 | |
| Regression Coefficient | Age | Academic | -0.012 | ** | -0.038 | *** |
| | | | (0.004) | | (0.001) | |
| | Gender | Academic | 0.334 (0.059) | *** | 0.229 (0.017) | *** |
| | Age | Learning Env | -0.005 (0.005) | | -0.018 (0.001) | *** |
| | Gender | Learning Env | 0.247 (0.073) | *** | 0.184 (0.019) | *** |
| | Age | Campus Culture | -0.011 (0.005) | * | -0.021 (0.001) | *** |
| | Gender | Campus Culture | 0.310 (0.078) | *** | 0.232 (0.012) | *** |
| | Age | Interpersonal | -0.012 (0.005) | * | -0.022 (0.001) | *** |
| | Gender | Interpersonal | 0.213 (0.077) | ** | 0.296 (0.014) | *** |
| | Age | Personal | -0.007 (0.005) | | -0.008 (0.001) | *** |
| | Gender | Personal | 0.271 (0.067) | *** | 0.253 (0.014) | *** |

(1) Model 1 fit statistics: $X^2$ = 4218, df = 1061, $p$<0.001, RMSEA = 0.076 (95% CI 0.074, 0.078), SRMR = 0.057, CFI = 0.810, TLI = 0.800.

(2) Model 2 fit statistics: $X^2$ = 92240, df = 1061, $p$<0.001, RMSEA = 0.085 (95% CI 0.084, 0.085), SRMR = 0.082, CFI = 0.640, TLI = 0.618.

(3) Regression coefficient estimates $\beta$ (Standard Error). Indicates statistical significance at

* $p$<0.05, ** $p$<0.01, *** $p$<0.001.

## Conclusion

The results of this research provide further evidence for the validity of the PSSI among varied university student populations in Canada and addresses several of the limitations identified in earlier preliminary psychometric work on the instrument. Future research will continue to explore the utility of the PSSI in identifying distributions of stressors among unique subgroups of interest (i.e., international students, first-generation students, etc.), as well as the continued validation of the instrument in different contexts (i.e., French-speaking students, college populations). Finally, the development of a shortened version of the PSSI to improve its accessibility and utility in terms of its ability to be added on to existing survey instruments will also be explored.

**Table 4.** MIMIC model estimates exploring relations to other variables ($n$ = 516).

| Indicators | Latent Variable | Estimate (SE) | $p$ |
|---|---|---|---|
| PSS10 | Academic | 0.019 (0.006) | 0.001 |
| K10 | | 0.028 (0.003) | <0.001 |
| PSS10 | Learning Env | 0.016 (0.008) | 0.041 |
| K10 | | 0.028 (0.004) | <0.001 |
| PSS10 | Campus Culture | 0.021 (0.008) | 0.007 |
| K10 | | 0.039 (0.004) | <0.001 |
| PSS10 | Interpersonal | 0.026 (0.007) | <0.001 |
| K10 | | 0.037 (0.004) | <0.001 |
| PSS10 | Personal | 0.026 (0.006) | <0.001 |
| K10 | | 0.033 (0.004) | <0.001 |

(1) Model fit statistics: RMSEA = 0.076 (95% CI 0.074, 0.078), SRMR = 0.057, CFI = 0.811, TLI = 0.799, $X^2$ = 4348, df = 1061, $p$<0.001.

(2) SE = Standard error

**Table 5. Relations to other variables (Spearman's rho correlations ($r_s$)) by group.**

| Group | PSSI and K10 | | | PSSI and PSS10 | | |
|---|---|---|---|---|---|---|
| | $r_s$ | $p$ | 95% CI | $r_s$ | $p$ | 95% CI |
| **Total Sample** | 0.47 | <0.001 | 0.45, 0.48 | 0.44 | <0.001 | 0.43, 0.46 |
| **Sex** | | | | | | |
| Female | 0.44 | <0.001 | 0.42, 0.45 | 0.40 | <0.001 | 0.38, 0.42 |
| Male | 0.51 | <0.001 | 0.48, 0.54 | 0.51 | <0.001 | 0.48, 0.53 |
| **Level of Study** | | | | | | |
| Undergraduate | 0.46 | <0.001 | 0.44, 0.48 | 0.44 | <0.001 | 0.42, 0.45 |
| Graduate | 0.46 | <0.001 | 0.42, 0.49 | 0.45 | <0.001 | 0.42, 0.48 |
| **Course Load** | | | | | | |
| Full-time | 0.46 | <0.001 | 0.44, 0.47 | 0.44 | <0.001 | 0.42, 0.45 |
| Part-time | 0.52 | <0.001 | 0.47, 0.57 | 0.49 | <0.001 | 0.44, 0.53 |
| **Status** | | | | | | |
| Domestic | 0.46 | <0.001 | 0.44, 0.48 | 0.45 | <0.001 | 0.43, 0.46 |
| International | 0.51 | <0.001 | 0.47, 0.55 | 0.45 | <0.001 | 0.41, 0.49 |

## Acknowledgments

BL would like to acknowledge and thank all the students who participated in this study, as well as each of the co-investigators at participating institutions who offered their time and commitment to this project during a challenging academic year.

## Author Contributions

**Conceptualization:** Brooke Linden, Heather Stuart.

**Data curation:** Brooke Linden.

**Formal analysis:** Brooke Linden.

**Investigation:** Brooke Linden.

**Methodology:** Brooke Linden, Heather Stuart.

**Project administration:** Brooke Linden.

**Software:** Brooke Linden.

**Supervision:** Heather Stuart.

**Validation:** Brooke Linden, Heather Stuart.

**Writing – original draft:** Brooke Linden.

**Writing – review & editing:** Brooke Linden, Heather Stuart.

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
