## [Decision Letter · Decision Letter 0]

2 Aug 2022

PONE-D-22-06413

Further evidence in support of the validity of the Post-Secondary Student Stressors Index using a nationwide, cross-sectional sample of Canadian university students

PLOS ONE

Dear Dr. Linden,

Thank you for submitting your manuscript to PLOS ONE. After careful consideration, we feel that it has merit but does not fully meet PLOS ONE’s publication criteria as it currently stands. Therefore, we invite you to submit a revised version of the manuscript that addresses the points raised during the review process.

Please note that we have only been able to secure a single reviewer to assess your manuscript. We are issuing a decision on your manuscript at this point to prevent further delays in the evaluation of your manuscript. Please be aware that the editor who handles your revised manuscript might find it necessary to invite additional reviewers to assess this work once the revised manuscript is submitted. However, we will aim to proceed on the basis of this single review if possible. 

The manuscript has been evaluated by one reviewer, and his comments are available below.

The reviewer has raised a number of concerns. He requests improvements to the reporting of methodological aspects of the study, for example, regarding the characteristics of the institutions and definition of post-secondary education. The reviewer also requests revision to discussion.

Could you please carefully revise the manuscript to address all comments raised?

We look forward to receiving your revised manuscript.

Kind regards,

Lorena Verduci

Staff Editor

PLOS ONE

https://journals.plos.org/plosone/s/file?id=ba62/PLOSOne_formatting_sample_title_authors_affiliations.pdf".

“I have read the journal's policy and the authors of this manuscript have the following competing interests: BL is the developer of the Post-Secondary Student Stressors Index (a non-financial conflict of interest). HS is the Bell Canada Mental Health and Anti-Stigma Research Chair.”

b) If there are no restrictions, please upload the minimal anonymized data set necessary to replicate your study findings as either Supporting Information files or to a stable, public repository and provide us

with the relevant URLs, DOIs, or accession numbers. For a list of acceptable repositories, please see http://journals.plos.org/plosone/s/data-availability#loc-recommended-repositories.

7. Your ethics statement should only appear in the Methods section of your manuscript. If your ethics statement is written in any section besides the Methods, please delete it from any other section.

Reviewers' comments:

Reviewer's Responses to Questions

**Comments to the Author**

1. Is the manuscript technically sound, and do the data support the conclusions?

Reviewer #1: Yes

2. Has the statistical analysis been performed appropriately and rigorously? 

Reviewer #1: Yes

3. Have the authors made all data underlying the findings in their manuscript fully available?

Reviewer #1: Yes

4. Is the manuscript presented in an intelligible fashion and written in standard English?

Reviewer #1: Yes

5. Review Comments to the Author

Reviewer #1: This manuscript is a continuation of previous studies that provides additional evidence to support the use of an indicator on stressors in students in Canada. With this, it seeks to identify the main sources of stressors and improve mental health services.

The creation of the instrument and its validation fills a gap well exposed by the researchers related to the fact that there is no such tool with the participatory approach that is proposed.

Major comments

1. Do the participating institutions have the same characteristics? I am referring to whether there may be different sources of stress depending on the profile of the participating institution. Perhaps it would help to mention that the type of institutions was comparable if they were.

2. Is there a difference between post-secondary education and higher education? I understand that it includes not only universities, but also other types of educational institutions. I recommend clarifying what you mean by post-secondary education.

3. According to the domains of stressors and their respective items, (although the development of the instrument should be in another publication) could you describe how these were chosen and how was this a participatory process?

4. I recommend getting more out of the results in the discussion. An important topic that could be included in the discussion is that when identifying the sources of stress, it comes to light that most of them have nothing to do with the quality and quantity of mental health services in institutions. Rather, the sources of stress come from, for example, scheduling, organization and evaluation methods, communication and relationship between teachers and students, or even sexual harassment on campus. How do the authors think these sources of stress can be addressed on campus? can they be addressed? is there an example of that?

Minor comments

References. Data presentations, spelling, grammar, reagent word.

1. I recommend changing the post-secondary keyword since it's in the title.

2. There is an error in the results since it says that the ages range from 16 to 20 years and in the table it is up to 30.

6. PLOS authors have the option to publish the peer review history of their article (what does this mean?). If published, this will include your full peer review and any attached files.

Reviewer #1: No

---

## [Author Response · Author response to Decision Letter 0]

20 Sep 2022

Response to Reviewers 

Manuscript #PONE-D-22-06413: “Further evidence in support of the validity of the Post-Secondary Student Stressors Index using a nationwide, cross-sectional sample of Canadian university students”

Thank you to the reviewers for the opportunity to revise this manuscript. Please find below point-by-point responses to your inquiries, comments, and suggestions.

Reviewer #1:

This manuscript is a continuation of previous studies that provides additional evidence to support the use of an indicator on stressors in students in Canada. With this, it seeks to identify the main sources of stressors and improve mental health services. The creation of the instrument and its validation fills a gap well exposed by the researchers related to the fact that there is no such tool with the participatory approach that is proposed.

Comment 1: Do the participating institutions have the same characteristics? I am referring to whether there may be different sources of stress depending on the profile of the participating institution. Perhaps it would help to mention that the type of institutions was comparable if they were.

Response: Thank you for this question. In fact, this was one of the questions we sought to answer through this cross-Canada, multi-site study, and indeed something that we are investigating in additional analyses. For example, we were interested in learning whether patterns of stress would appear to be different based on region in Canada, size of campus, and nature of campus (with respect to them being “commuter” campuses, where students typically live off campus and drive in to attend school vs. non-commuter campuses, where the vast majority of students live on or very near campus). These are all research questions that are being answered in additional sub-studies of this data, as they are beyond the scope of this particular paper. 

For the purposes of this paper, we sought to investigate whether the psychometric properties of the index would remain stable and similar to what we observed in our pilot study at a single, Eastern Ontario university. In fact, we note that a major goal of the study was to collect further evidence of validity using a larger, more varied sample (line 66-68) – therefore the strong psychometrics our analyses revealed using this sample given the hypothesized variation in institutions’ characteristics lends additional strength to our findings.

Comment 2: Is there a difference between post-secondary education and higher education? I understand that it includes not only universities, but also other types of educational institutions. I recommend clarifying what you mean by post-secondary education.

Response: No, “post-secondary education” and “higher education” are synonymous. The PSSI was developed and intended to be used across different types of higher education institutions, including universities and colleges (hence “post-secondary stressors index”, and not “university stressors index”). However, the reviewer has likely noted that the majority of the participating institutions in this sample were universities (the one exception being Aurora College). Admittedly, this is a limitation of the tool as it has not yet been widely tested among college populations (this is in the works). 

In response to this comment, we have now reviewed all instances throughout the manuscript where we have referred to “post-secondary” and adjusted this to “university” as appropriate (note that the word university is used when we refer to our sample or the applicability of the PSSI, but when referencing general mental health statistics for post-secondary student populations, for example, the word post-secondary is used as these stats apply to both university and college populations. We have also added a statement in our limitations section regarding the lack of testing to date among college populations potentially limiting the generalizability of the PSSI in that context (lines 283-286).

Comment 3: According to the domains of stressors and their respective items, (although the development of the instrument should be in another publication) could you describe how these were chosen and how was this a participatory process?

Response: The reviewer is correct that details regarding the development of the PSSI have been published elsewhere (please see Linden & Stuart 2019 [doi: 10.1186/s12889-019-7472-z] and Linden, Boyes & Stuart 2020 [doi: 10.1080/07448481.2020.1754222]). Briefly, the PSSI was developed through detailed consultation with students, including the selection and naming of the domains. Naming of the domains was decided over the course of multiple focus group interviews with diverse groups of students. 

Comment 4: I recommend getting more out of the results in the discussion. An important topic that could be included in the discussion is that when identifying the sources of stress, it comes to light that most of them have nothing to do with the quality and quantity of mental health services in institutions. Rather, the sources of stress come from, for example, scheduling, organization and evaluation methods, communication and relationship between teachers and students, or even sexual harassment on campus. How do the authors think these sources of stress can be addressed on campus? can they be addressed? is there an example of that?

Response: With respect to this component of the author’s statement: “…when identifying the sources of stress, it comes to light that most of them have nothing to do with the quality and quantity of mental health services in institutions”, we are unable to comment as we did not do any analytic work that might provide support for a relationship (or lack of relationship) between he quality/quantity of mental health services provided on campuses and the stressors students are experiencing. While we appreciate the reviewer’s recommendation to add more discussion around the specific items that students identified as impactful sources of stress, we feel that would be beyond the scope of the present paper, which aims only to provide a concise assessment of the instrument’s psychometric properties. In fact, we have recently published an alternative paper that goes into detail regarding this exact question, exploring the stressors identified by students as impactful across each of the domains, as well as looking at whether trends appeared over time. We invite the reviewer to refer to Linden, Stuart, and Ecclestone 2022 (doi: 10.1177/07067437221111365). 

Comment 5: I recommend changing the post-secondary keyword since it's in the title.

Response: This keyword has now been changed to “university”.

Comment 6: There is an error in the results since it says that the ages range from 16 to 20 years and in the table it is up to 30.

Response: If the reviewer is referring to lines 167-168 in the “Sample” subsection under results, we state that the majority of respondents fell between the ages of 16 and 20, not all of the respondents (41.4% of our sample was under the age of 21, while the other age categories account for considerable smaller proportions of the sample). For clarity we have adjusted this sentence to read “Most participants were aged 20 years or younger…” to better align with the age categories as displayed in Table 1. Please note that in reviewing this paragraph, we did notice some inconsistencies in the proportions reported in text - these have all now been fixed to align with the numbers in the table (lines 164-169).

---

## [Decision Letter · Decision Letter 1]

14 Nov 2022

PONE-D-22-06413R1Further evidence in support of the validity of the Post-Secondary Student Stressors Index using a nationwide, cross-sectional sample of Canadian university studentsPLOS ONE

Dear Dr. Linden,

Thank you for submitting your manuscript to PLOS ONE. After careful consideration, we feel that it has merit but does not fully meet PLOS ONE’s publication criteria as it currently stands. Therefore, we invite you to submit a revised version of the manuscript that addresses the points raised during the review process.

We look forward to receiving your revised manuscript.

Kind regards,

Supat Chupradit, Ph.D., M.Ed., B.Sc.(OT), B.P.A., B.Ed., B.A.

Academic Editor

PLOS ONE

Journal Requirements:

Reviewers' comments:

Reviewer's Responses to Questions

**Comments to the Author**

1. If the authors have adequately addressed your comments raised in a previous round of review and you feel that this manuscript is now acceptable for publication, you may indicate that here to bypass the “Comments to the Author” section, enter your conflict of interest statement in the “Confidential to Editor” section, and submit your "Accept" recommendation.

Reviewer #1: All comments have been addressed

Reviewer #2: All comments have been addressed

Reviewer #3: All comments have been addressed

Reviewer #4: All comments have been addressed

2. Is the manuscript technically sound, and do the data support the conclusions?

Reviewer #1: Yes

Reviewer #2: Yes

Reviewer #3: Yes

Reviewer #4: Yes

3. Has the statistical analysis been performed appropriately and rigorously? 

Reviewer #1: Yes

Reviewer #2: Yes

Reviewer #3: Yes

Reviewer #4: Yes

4. Have the authors made all data underlying the findings in their manuscript fully available?

Reviewer #1: Yes

Reviewer #2: Yes

Reviewer #3: Yes

Reviewer #4: Yes

5. Is the manuscript presented in an intelligible fashion and written in standard English?

Reviewer #1: Yes

Reviewer #2: Yes

Reviewer #3: Yes

Reviewer #4: Yes

6. Review Comments to the Author

Reviewer #1: I have reviewed the responses given by the authors to my comments. I am satisfied with the answers and recommend the publication of this manuscript.

Reviewer #2: Overall, this paper is quite interesting and has been revised based on the feedback of a previous reviewer. However, there are some areas that need improvement. The Abstract should be written in one paragraph only. The demographic data of the samples should be mentioned in the Research method, not the Research results.

Reviewer #3: Further evidence in support of the validity of the Post-Secondary Student Stressors Index using a nationwide, cross-sectional sample of Canadian university students. This manuscript revision, I'm accept all comments that author response. Accept

Reviewer #4: Thank you for considering my comments and suggestions. I am satisfied with the responses.

All the best for your article.

7. PLOS authors have the option to publish the peer review history of their article (what does this mean?). If published, this will include your full peer review and any attached files.

Reviewer #1: **Yes: **Mónica Suárez-Reyes

Reviewer #2: **Yes: **Kittisak JERMSITTIPARSERT

Reviewer #3: No

Reviewer #4: No

---

## [Author Response · Author response to Decision Letter 1]

14 Nov 2022

Response to Reviewers 

Manuscript #PONE-D-22-06413: “Further evidence in support of the validity of the Post-Secondary Student Stressors Index using a nationwide, cross-sectional sample of Canadian university students”

Thank you to the reviewers for the opportunity to revise this manuscript. Please find below point-by-point responses to your inquiries, comments, and suggestions.

Journal Requirements:

Comment 1: Please review your reference list to ensure that it is complete and correct. If you have cited papers that have been retracted, please include the rationale for doing so in the manuscript text, or remove these references and replace them with relevant current references. Any changes to the reference list should be mentioned in the rebuttal letter that accompanies your revised manuscript. If you need to cite a retracted article, indicate the article’s retracted status in the References list and also include a citation and full reference for the retraction notice.

Response: The reference list (and in-text citations) have now been reviewed and are correct.

Reviewer Comments:

Comment 1: The Abstract should be written in one paragraph only. 

Response: We have now shortened the abstract to be the length of a single paragraph.

Comment 2: The demographic data of the samples should be mentioned in the Research method, not the Research results.

Response: We have elected not to make this requested change as we feel it is more appropriate to have the demographic statistical results reported in the Results section with all other results.

---

## [Decision Letter · Decision Letter 2]

25 Nov 2022

Further evidence in support of the validity of the Post-Secondary Student Stressors Index using a nationwide, cross-sectional sample of Canadian university students

PONE-D-22-06413R2

Dear Dr. Linden,

We’re pleased to inform you that your manuscript has been judged scientifically suitable for publication and will be formally accepted for publication once it meets all outstanding technical requirements.

Kind regards,

Supat Chupradit, Ph.D., M.Ed., B.Sc.(OT), B.P.A., B.Ed., B.A.

Academic Editor

PLOS ONE

Additional Editor Comments (optional):

Reviewers' comments:

Reviewer's Responses to Questions

**Comments to the Author**

1. If the authors have adequately addressed your comments raised in a previous round of review and you feel that this manuscript is now acceptable for publication, you may indicate that here to bypass the “Comments to the Author” section, enter your conflict of interest statement in the “Confidential to Editor” section, and submit your "Accept" recommendation.

Reviewer #1: All comments have been addressed

Reviewer #3: All comments have been addressed

Reviewer #4: All comments have been addressed

2. Is the manuscript technically sound, and do the data support the conclusions?

Reviewer #1: Yes

Reviewer #3: Yes

Reviewer #4: Yes

3. Has the statistical analysis been performed appropriately and rigorously? 

Reviewer #1: Yes

Reviewer #3: Yes

Reviewer #4: Yes

4. Have the authors made all data underlying the findings in their manuscript fully available?

Reviewer #1: Yes

Reviewer #3: Yes

Reviewer #4: Yes

5. Is the manuscript presented in an intelligible fashion and written in standard English?

Reviewer #1: Yes

Reviewer #3: Yes

Reviewer #4: Yes

6. Review Comments to the Author

Reviewer #1: The authors have responded to my comments satisfactorily. The article is interesting and provides relevant information on the study of health promotion in the university context. I recommend accepting the article.

Reviewer #3: Further evidence in support of the validity of the Post-Secondary Student Stressors Index using a nationwide, cross-sectional sample of Canadian university students. Overall improve by reviewer's comments. accept.

Reviewer #4: Further evidence in support of the validity of the Post-Secondary Student Stressors Index using a nationwide, cross-sectional sample of Canadian university students. I approve this manuscript that revise version.

7. PLOS authors have the option to publish the peer review history of their article (what does this mean?). If published, this will include your full peer review and any attached files.

Reviewer #1: **Yes: **Mónica Suárez-Reyes

Reviewer #3: No

Reviewer #4: No

---

## [Editor Report · Acceptance letter]

7 Dec 2022

PONE-D-22-06413R2 

Further evidence in support of the validity of the Post-Secondary Student Stressors Index using a nationwide, cross-sectional sample of Canadian university students 

Dear Dr. Linden:

I'm pleased to inform you that your manuscript has been deemed suitable for publication in PLOS ONE. Congratulations! Your manuscript is now with our production department. 

Kind regards, 

on behalf of

Assistant Professor Supat Chupradit 

Academic Editor

PLOS ONE